# A Putative Lignin Copper Oxidase from *Trichoderma reesei*

**DOI:** 10.3390/jof7080643

**Published:** 2021-08-07

**Authors:** Mariane Daou, Alexandra Bisotto, Mireille Haon, Lydie Oliveira Correia, Betty Cottyn, Elodie Drula, Soňa Garajová, Emmanuel Bertrand, Eric Record, David Navarro, Sana Raouche, Stéphanie Baumberger, Craig B. Faulds

**Affiliations:** 1BBF, INRAE, Aix Marseille University, 13288 Marseille, France; mariane.daou@ku.ac.ae (M.D.); alexandra.bisotto@gmail.com (A.B.); mireille.haon@inrae.fr (M.H.); elodie.drula@inrae.fr (E.D.); sona.garajova@univ-amu.fr (S.G.); emmanuel.bertrand@univ-amu.fr (E.B.); eric.record@inrae.fr (E.R.); david.navarro@inrae.fr (D.N.); sana.raouche@univ-amu.fr (S.R.); 2PAPPSO Platform, INRAE, AgroParisTech, Micalis Institute, Université Paris-Saclay, 78350 Jouy-en-Josas, France; lydie.oliveira-correia@inrae.fr; 3Institut Jean-Pierre Bourgin, INRAE, AgroParisTech, Université Paris-Saclay, 78000 Versailles, France; betty.cottyn@inrae.fr (B.C.); stephanie.baumberger@inrae.fr (S.B.); 4CIRM-CF BBF, INRAE, Aix Marseille University, 13288 Marseille, France

**Keywords:** *Trichoderma reesei*, technical lignin, copper radical oxidase

## Abstract

The ability of *Trichoderma reesei*, a fungus widely used for the commercial production of hemicellulases and cellulases, to grow and modify technical soda lignin was investigated. By quantifying fungal genomic DNA, *T. reesei* showed growth and sporulation in solid and liquid cultures containing lignin alone. The analysis of released soluble lignin and residual insoluble lignin was indicative of enzymatic oxidative conversion of phenolic lignin side chains and the modification of lignin structure by cleaving the β-O-4 linkages. The results also showed that polymerization reactions were taking place. A proteomic analysis conducted to investigate secreted proteins at days 3, 7, and 14 of growth revealed the presence of five auxiliary activity (AA) enzymes in the secretome: AA6, AA9, two AA3 enzymes), and the only copper radical oxidase encoded in the genome of *T. reesei*. This enzyme was heterologously produced and characterized, and its activity on lignin-derived molecules was investigated. Phylogenetic characterization demonstrated that this enzyme belonged to the AA5_1 family, which includes characterized glyoxal oxidases. However, the enzyme displayed overlapping physicochemical and catalytic properties across the AA5 family. The enzyme was remarkably stable at high pH and oxidized both, alcohols and aldehydes with preference to the alcohol group. It was also active on lignin-derived phenolic molecules as well as simple carbohydrates. HPSEC and LC-MS analyses on the reactions of the produced protein on lignin dimers (SS ββ, SS βO4 and GG β5) uncovered the polymerizing activity of this enzyme, which was accordingly named lignin copper oxidase (*Tr*LOx). Polymers of up 10 units were formed by hydroxy group oxidation and radical formation. The activations of lignin molecules by *Tr*LOx along with the co-secretion of this enzyme with reductases and FAD flavoproteins oxidoreductases during growth on lignin suggest a synergistic mechanism for lignin breakdown.

## 1. Introduction

Agaricomycetes have been long studied for their ability to mineralize woody biomass. A number of representative white-rot fungi belonging to the Basidiomycetes, such as *Phanerochaete chrysosporium*, *Trametes versicolor*, and *Pleurotus ostreatus*, have in particular been analyzed because of their effectiveness in converting lignocellulosic material into CO_2_, primarily due to the enzymes (see [1], 2015, and references therein) and small organic molecules that they secrete during growth on such biomass. There is continual interest in obtaining fermentable sugars [2,3,4], high value animal feed [5], and high value chemical precursors to replace fossil-fuel-derived compounds through environmental processes, and the initiative towards zero waste within a circular bioeconomy framework [6]. This continues to drive towards understanding further how secreted microbial enzymes interact with their substrates and each other, and what factors prevent or enhance such interactions.

The impact of the connectivity of the individual components within woody and non-woody lignocellulosic feedstocks on their biodegradability has to be understood, and the use of proteomic, transcriptomic and secretomic tools to understand the microbial enzymatic mechanisms is greatly expanding this knowledge [7,8,9]. Many recently published works have concentrated on these-Omic approaches coupled with biochemical and microbiological analysis for the fungal degradation of the lignocellulosic components from divergent plant sources [7,8,10,11,12,13,14]. The arsenal of enzymes that these fungi secrete at different time points of growth are still largely unknown, but their presence leads to an empirical improved boost in saccharification for the production of biofuels or the release of small molecules of interest for further bioconversion. Changes to the composition of the biomass have facilitated the study of enzyme synergies and approaches for determining the systematic order of biomass deconstruction, and especially the untangling of the lignin component from the polysaccharides [15]. However, due to the recalcitrant nature of the lignin component, coupled with obtaining a well-characterized native lignin through chemical extraction, less is known about how fungi specifically degrade lignin, either native or technical from industrial processes, compared to that from the more composite lignocellulose. 

Investigations into a successive degradation pattern of wood colonization by fungi have illustrated that the Basidiomycetes dominate the initial stages of deconstruction of the biomass, while Ascomycetes take over this role in the later stages [16,17]. Ascomycetes fungi are generally regarded as soft-rot degraders of various forms of plant biomass, and compared to the basidiomycetes, their ability to degrade lignin is considered as being limited [18,19]. Soft-rot fungi are found on a wide array of lignocellulosic substrates, such as herbaceous plant debris (wheat straw), whereas basidiomycetous decayers are usually restricted to wood [20]. One possibility suggested is that ascomycetes have a limited arsenal of lignin-attacking enzymes, and so are capable of attacking only the phenolic units in lignin which are chemically more labile than the remaining ether-linked, nonphenolic units [20,21], which require action of stronger oxidants, such as the oxyradicals produced by ligninolytic peroxidases or laccases in the presence of various redox mediators. In 2007, Shary and co-workers demonstrated that the ascomycete *Daldinia concentrica* produced a phenol oxidase activity that could act directly on such phenol units, and proposed that this cleavage of the C_α_-C_β_ propyl side chains was one of the major mechanisms of this fungus being able to degrade lignin [22]. Another suggestion for the production of these oxidases is the natural production of H_2_O_2_ to act as a bactericide in the environment [23]. The coprophilous ascomycete, *Podospora anserina*, more known for its ability to degrade carbohydrate, has been one ascomycete studied for its ligninolytic activities due to its ease of culture and genetic analysis. This fungus has been shown to produce laccases, which are involved in phenolic compound oxidation, although a role in actual lignin depolymerization was inconclusive [24], and a catalase capable of harnessing released hydrogen peroxide [25]. The fungus has also been shown to degrade lignin through the reduction of β-O-4′ aryl ether linkages, a linkage that represents more than 50% of all inter-unit linkages of native lignins (softwood: 45–50%, hardwood: 60–80%, grasses: 69–94%) [26,27,28]. Previously, an extracellular β-etherase capable of cleaving this bond was isolated from an ascomycete of the genus *Chaetomium* [29]. Furthermore, the released phenolics further stimulated growth of *P. anserina* due to augmented radical oxygen species production [27]. 

As a component of the secreted enzymatic arsenal utilized by fungi, the copper radical oxidases (CRO, previously also known as radical-copper oxidases; [30]) have generally been regarded as an accessory enzyme in the ligninolysis process, generating H_2_O_2_ for the heavy cleavage action of peroxidases [31]. They have subsequently been found to require themselves the presence of a peroxidase or catalase in a two-fold mechanistic function: (1) to utilize the H_2_O_2_ produced during catalysis and thus avoid auto-inactivation, and (2) for full activation of the resting state of the CRO [32,33,34]. The genes encoding CROs are widely distributed in the Fungal Kingdom, and the copper-binding catalytic center is well preserved. Five CRO subfamilies have been recognized for Basidiomycetes [35,36], all belonging to the AA5_1 family of the CAZy classification [37,38], and clearly separate from the galactose oxidase (GalOx) clades in AA5_2 [39], although recent studies are showing the existence of AA5-classified oxidase from the Ascomycete *Penicillium rubens* with a functional overlap between the two subfamilies [40]. Glyoxal oxidase (GLOX) has been the most extensively studied of the AA5_1 CROs to date [32,36,41], with their ability to utilize an array of aldehydes and α-hydroxycarbonyls as the substrates converting them in sequential reactions to carboxylic acids [35,36].

During the published study on three basidiomycetes grown on a technical soda lignin, Protobind 1000, a number of enzymes were identified in the secretomes specific to growth of the fungi on lignin alone, including CROs [42]. As part of the same study, the ascomycete *Trichoderma reesei* was also examined, initially as an outlier as it is known as a cellulolytic fungus rather than a lignocellulolytic one [43]. This paper describes the identification, cloning, heterologous expression of a CRO identified in the secretome, together with an investigation into its substrate specificity. The relation of this enzyme to the residual technical soda lignin after growth of *T. reesei* is also discussed.

## 2. Materials and Methods

### 2.1. Fungal Strain 

Polyploid strain of the fungus *Trichoderma reseei* BRFM 1104 (QM6a) was obtained from the International Centre of Microbial Resources, Marseille, France (CIRM-CF; https://www.cirm-fungi.fr, accessed 7 August 2021). The identity of the strain was checked by morphological observations and molecular analysis of Internal Transcribed Spacer sequences compared to Genbank [44]. The strain was maintained on potato dextrose agar (Sigma-Aldrich, St. Louis, MO, USA) slants at 4 °C.

### 2.2. Growth on Lignin 

The soda technical lignin (Protobind 1000) used in this study was produced from a wheat straw and Sarkanda grass mix and purchased from GreenValue Enterprises LLC (Media, PA, USA). The sample was analyzed for Klason lignin (88.1%), carbohydrates (1.9%, of which 1.2% xylose, 0.3% arabinose, 0.1% galactose and 0.2% glucose), free phenolic monomers (1.4%) and ash contents (1.4%) [42]. 

From agar slants, *T. reseei* was transferred and grown on potato dextrose agar plates (Sigma-Aldrich, St. Louis, MO, USA). Cultures on minimal media plates containing 10 g/L technical soda lignin and in submerged cultures containing 100 mL of lignin-containing media (10 g/L technical soda lignin) were performed as previously described [42]. Control conditions were supplemented with 1 g/L glucose and 2.5 g/L maltose for plate and liquid cultures, respectively. All cultures were performed in duplicate.

Fungal growth on plates was followed by measuring the radial expansion of the fungus (cm/day). Growth in liquid cultures was followed by quantifying the fungal DNA material and relating it to the mycelial dry weight as previously described [42,45].

Residual solid material from liquid cultures was harvested and used for the characterization of insoluble lignin. Concentrated supernatants were used for proteomic analysis. The flow through from the concentration step was also collected to analyze changes in the water-soluble components of technical soda lignin.

### 2.3. Characterization of Residual Lignin

Ten milliliters (10 mL) of culture supernatant from day 14 of growth and the control were used to analyze soluble lignin residues by High Pressure Size Exclusion chromatography method (HPSEC) (Dionex Ultimate 3000, Thermo Scientific, Saint Aubin, France) and liquid chromatography–mass spectrometry (LC–MS) (UHPLC, Thermo Scientific; Impact II, Bruker, Billerica, MA, USA), after extraction with ethyl acetate. The control consisted of the culture media containing lignin in the absence of fungi and was incubated and recovered under the same conditions.

Residual technical soda lignin PB1000 was recovered after fungal growth by the centrifugation of the culture medium on day 14. The insoluble fraction was analyzed by HPSEC and quantitative ^31^P NMR (Ascend 400 MHz Spectrometer, Bruker). The thioacidolysis monomers p-hydroxyphenyl (H), guaiacyl (G), and syringyl (S) were also analyzed as their trimethylsilyl derivatives by gas chromatography−mass spectrometry (GC–MS, Saturn 2100, Varian, Palo Alto, CA, USA) as previously described [46]. The GC-MS apparatus was equipped with a poly (dimethylsiloxane) capillary column (30 m × 0.25 mm; SPB-1, Supelco, Bellefonte, PA, USA) operating in the temperature program (40 to 180 °C at 30 °C/min, then 180 to 260 °C at 2 °C/min). The determination of the thioethylated H, G, and S monomers was performed from ion chromatograms reconstructed at *m*/*z* 239, 269, and 299, respectively, as compared to the internal standard (heneicosane) signal measured from the ion chromatogram reconstructed at *m*/*z* (57 + 71 + 85).

The detailed description of the analysis methods and the parameters used have been previously described [42]. 

### 2.4. Proteomic Analysis

The proteomic analysis was performed as previously described [42] at PAPPSO platform facilities (http://pappso.inra.fr/, accessed 7 August 2021) for protein identification. LC–MS/MS analyses were performed using a NanoLC Ultra system (Eksigent, Dublin, CA, USA) connected to a Q-Exactive Plus mass spectrometer (Thermo Fisher Scientific, Waltham, MA, USA) and an Ultimate 3000 RSLC system (Thermo Fisher Scientific) coupled to an LTQ-orbitrap discovery mass spectrometer (Thermo Fisher Scientific) by nanoelectrospray ion source on both systems.

All MS/MS spectra were integrated against the JGI databases for *T. reseei* v2.0 (https://genome.jgi.doe.gov/Trire2/Trire2.home.html, accessed 7 August 2021) using the X!TandemPipeline (X!Tandem version 3.4.3), the open search engine developed by PAPPSO (http://pappso.inra.fr/bioinfo/xtandempipeline/, accessed 7 August 2021). Data filtering was achieved according to a peptide *E*-value < 10^−2^, protein *E*-value < 10^−3^ and to a minimum of two identified peptides per protein. As some of the proteins were only detected by one of the used methods, identifications from both mass spectrometers were combined.

All amino acid sequences were obtained from JGI Mycocosm [47].

### 2.5. Phylogenetic Study and Sequence Alignments

The sequences selected to build this tree are public accessions from NCBI (https://www.ncbi.nlm.nih.gov/, accessed 7 August 2021) and JGI. (https://mycocosm.jgi.doe.gov/mycocosm/home, accessed 7 August 2021). These were cleaned of their peptide signal and WSC domain(s) (cell wall integrity and stress response components) when present. The multi-copy sequences from a single organism have been removed to avoid redundancy. Only one representative sequence was included. A multiple alignment was performed using MAFFT tool (version 7) following the accuracy-oriented method and the option maxiterate 1000 input [48]. The quality of the alignment was confirmed by the transitive consistency score [49]. TrimAl v1.2 was used to automatically remove spurious or misaligned sequences with the automated option [50]. The phylogenetic tree was produced from the NGphylogeny [51] and PhyML [52] (version 3.0) and formatted using iTOL [53]. Lignin copper oxidase (*Tr*LOx) protein domains were predicted using NCBI Conserved Domain Search [54]. The molecular structure was modelled using Phyre2 [55] and the images were generated using Pymol Molecular Graphics System (Version 2.0, Schrödinger, LLC, New York, NY, USA). 

### 2.6. Production of TrLOx in Pichia pastoris

The sequence encoding *Tr*LOx (GenBank accession number: MZ436831) was previously identified in the genome of *T. reesei* [56]. The native signal peptide sequence was removed (MKPSPVASLLSVSLLSLTSCHA), and the cDNA sequence was codon optimized for *P. pastoris* expression and synthesized (Genewiz, Leipzig, Germany). The synthetic gene was cloned in pPICZ alpha A (Invitrogen, Cergy-Pontoise, France) vector in the XhoI/XbaI site in frame with both the α-factor and the His6 tag at the C terminus of the recombinant protein. *Escherichia coli* strain DH5α (Invitrogen) was used for vector storage and propagation. 

pPICZαA recombinant plasmids were linearized with PmeI and used to transform competent *P. pastoris* SuperMan_5_ cells (BioGrammatics, Carlsbad, CA, USA) by electroporation. Zeocin-resistant *P. pastoris* transformants were then screened for protein production. Electrocompetent cells preparation, electroporation and screening were carried out as previously described [57] and proteins were analyzed by SDS-PAGE. 

The best-producing transformant was grown in 2 L buffered complex glycerol (BMGY) medium (10 g/L glycerol, 10 g/L yeast extract, 20 g/L peptone, 3.4 g/L yeast nitrogen base (YNB), 10 g/L ammonium sulfate, 100 mM phosphate buffer pH 6 and 0.2 g/L of biotin) at 30 °C and 200 rpm to an optical density at 600 nm of 2–6. Cultures were then centrifuged at 6000 rpm for 5 min and the pellet was dissolved in 100 mL buffered methanol-complex (BMMY) medium (10 g/L yeast extract, 20 g/L peptone, 3.4 g/L YNB, 10 g/L ammonium sulfate, 100 mM phosphate buffer pH 6 and 0.2 g/L of biotin) supplemented with 1 mL/L *Pichia* trace minerals 4 (PTM4) salt solution. Protein production was induced at 20 °C and 200 rpm by adding 3% methanol (*v*/*v*) daily for 3 days. The culture supernatant was collected by centrifugation (4000 rpm for 10 min at 4 °C). 

The preparation of the used media is described in the manufacturer’s manual (Invitrogen).

### 2.7. Protein Purification

The pH of the collected culture supernatant was adjusted to 7.8 with NaOH (1 M) and sterile filtered (pore size; 0.22 m; Express Plus; Merck Millipore, Guyancourt, France). Immobilized metal affinity chromatography (IMAC) using an Äkta purifier (GE Healthcare Life Sciences, Buc, France) was used for the purification of His-tagged *Tr*LOx protein. The sample was loaded onto a 5 mL HisTrap HP column prepacked with Ni Sepharose and equilibrated with binding buffer (50 mM Tris-HCl, pH 7.8, 150 mM NaCl). The proteins were eluted with 30% elution buffer (50 mM Tris-HCl, pH 7.8, 150 mM NaCl, 500 mM imidazole). Recovered recombinant proteins were concentrated using 10 kDa vivaspin concentrator (Sartorius, Aubagne, France) then dialyzed against 50 mM sodium phosphate, pH 7.

### 2.8. Protein Characterization

The concentration of the purified proteins was determined spectrophotometrically at 280 nm (ε = 193,513 M^−1^·cm^−1^) using a NanoDrop 2000 spectrophotometer (Thermo Fisher Scientific). Purified proteins were loaded onto 12% SDS-polyacrylamide gel, which was then stained with Coomassie blue, and the molecular mass of the protein was estimated according to the standard markers PageRuler prestained protein ladder (10 to 180 kDa; Thermo Fisher Scientific).

### 2.9. Enzyme Activity

The activity of *Tr*LOx was measured spectrophotometrically by detecting the H_2_O_2_-dependent oxidation of ABTS (Sigma-Aldrich) by horseradish peroxidase (HRP; Sigma-Aldrich) [36,58,59]. The reaction mixture contained 50 mM sodium tartrate buffer, pH 6, 7 units of type II HRP, 0.1 mM ABTS, 173 nM *Tr*LOx enzyme and 5 mM of substrate. The reaction was initiated by the addition of the substrate, and the activity was followed spectrophotometrically at 420 nm and 30 °C for 2 min. *Tr*LOx activity was tested on 70 different substrates purchased from Sigma-Aldrich including aromatic compounds, phenolics, aldehydes, ketones, carboxylic acids, furans, saccharides, lactones and alcohols (Table 1).

The effect of the metal ions Cu^2+^, Mg^2+^, Ni^2+^, Co^2+^, Mn^2+^, Fe^2+^ and Zn^2+^ was determined by measuring enzyme activity after pre-incubating the enzyme with 1 mM or 10 mM of metal for 0.5 h, 4 h and 24 h at 25 °C and 450 rpm shaking. The effects of the chelating agent EDTA and the inhibitor sodium azide were determined in a similar way. 

*Tr*LOx stability in H_2_O_2_ was investigated by incubating the enzyme with different concentrations (2, 4, 6, 8 and 10 mM) of H_2_O_2_ over different time periods (2, 4 and 24 h) at 4 °C. H_2_O_2_ was removed before adding the enzyme to the reaction mixture by washing the samples with buffer in NanoSEP OMEGA membrane 10 kDa centrifugal devices (PALL, Saint-Germain-en-Laye, France). 

In both assays, the standard activity assay was followed with 10 mM dihydroxyacetone as substrate. Enzyme activity is given as nanokatals per milligram of protein used in the reaction (nkat/mg), where 1 kat is the conversion rate of 1 mol of substrate per second. Relative activity was calculated as percentage of the activity in the absence of the tested compounds.

### 2.10. Steady-State Kinetics

The kinetic constants for *Tr*LOx were determined following the standard activity test and using dihydroxyacetone (0.005–100 mM), methylglyoxal (0.3–50 mM), galactose (0.3–1000 mM), glycerol (0.3–1000 mM), furfuryl alcohol (1–100 mM) and 2-phenylethanol (6–50 mM) as substrates. The Michaelis-Menten plots were used to calculate the kinetic parameters using OriginPro, Version 2020b (OriginLab Corporation, Northampton, MA, USA).

The generated H_2_O_2_ during *Tr*LOx activity on the selected substrates was also measured in a coupled reaction with HRP/Amplex Red (Thermo Fisher Scientific). The reaction was carried out as described previously with modifications [60]. A total volume of 100 µL contained 100 mM tartrate buffer pH 6, 50 µM Amplex Red, 7.1 U/mL HRP, 173 nM *Tr*LOx and 1 mM substrate. The fluorescence was followed at an excitation wavelength of 560 nm and an emission wavelength of 595 nm using a Tecan Infinite M200 plate reader (Tecan, Männedorf, Switzerland). The slope from the standard curve relating H_2_O_2_ concentration and fluorescence was used to calculate the amount of generated H_2_O_2_ over time (0–20 µM H_2_O_2_; 374.34 counts/µmol).

### 2.11. Temperature and pH Effect

The optimum temperature was determined against dihydroxyacetone from 20 °C to 80 °C, with 5 °C increments. The pH optimum was measured in sodium tartrate and sodium phosphate buffers over a pH range of 2 to 6 and 6 to 8, respectively, in 0.5 pH unit increments. 

The thermal stability was determined by incubating the enzyme at 30, 40, 50, 60, and 70 °C for 0.5, 1, 2, 4, 6, 8, 24 and 48 h. The enzyme was cooled on ice for 5 min before measurement. 

Similarly, the pH stability was analyzed by incubating the enzyme at 4 °C for 1, 2, 4, 6, 8, 24 and 48 h in sodium tartrate and sodium phosphate buffers in a pH range of 2 to 6 and 6 to 8, respectively, in 0.5 pH unit increments.

The residual activities were calculated as percentage of the measured activity before incubation. 

### 2.12. Activity of TrLOx on Lignin Derivatives

The activity of *Tr*LOx on the lignin-derived molecules syringol and syringyl alcohol was investigated by LC-MS. The reaction mixtures containing 10 mM of the substrate were incubated for 24 h at 30 °C and 850 rpm shaking and filtered (0.45 µm, GHP Acrodisc, Pall Gelman, Port Washington, NY, USA) before injection analysis on UltiMate 3000 LC system (Thermo Fisher Scientific) combined with an electrospray ionization mass spectrometer (ISQ-EM, Thermo Fisher Scientific). Analysis was performed using BEH C18 column (particle size 1.9 µm, length 150 mm, Waters, Milford, MA, USA). The samples were loaded at 0.2 mL/min and 30 °C. Used solvents were 0.2% formic acid (solvent A) and 100% acetonitrile (solvent B). Separation was achieved by multi-step gradient from 5–80% solvent B in 52 min. Negative and positive ion ESI–MS spectra (10–2000 *m*/*z*) were acquired (vaporizer temperature 61 °C, ion transfer temperature 300 °C, sheath gas pressure 21.8 psig, Auxiliary gas pressure 2.4 psig, sweep gas pressure 0.5 psig). The peaks were assigned according to the mass of the deprotonated (negative mode) and protonated (positive mode) ions and fragmentation pattern, and to the theoretical masses expected after the injection of pure commercial compounds.

The activity of *Tr*LOx on three lignin dimers was assessed by HPSEC. The reactions were carried out for 24 h at 30 °C with 850 rpm shaking and 3 different lignin dimers were used as substrates: BMA32 (SS βO4), AMA170 (SS syringaresinol) and AMA181 (GG β5). The dimers were synthesized using previously described methods [61] for dimers BMA32, AMA170 and [62] for dimer AMA181). The description of the HPSEC and LC-MS analysis and the parameters used have been previously detailed [42].

## 3. Results and Discussion

### 3.1. T. reesei Growth on Lignin

The ability of *T. reseei* BRFM 1104 to grow on 10 g/L soda lignin derived from wheat straw was investigated over a 14-day period. Fungal growth was observed on lignin-containing plates and the average radial expansion on this substrate was 0.62 cm/day (Appendix A). Supplementing the plates with glucose increased the radial expansion rate to 0.89 cm/day and enhanced sporulation (Appendix A). However, the typical green spores were also observed on lignin-alone-containing plates.

The growth in liquid cultures was measured by extracting and quantifying fungal genomic DNA. Although growth was greater in maltose-supplied medium, *T. reesei* was still able to grow on lignin alone with maximum growth reached after day 3 and the total quantified fungal genomic DNA was found to decrease overtime in both conditions (Appendix A). This can be explained by cell death and DNA denaturation in the minimal culture medium containing only lignin. 

The growth and sporulation in the absence of carbon source other than lignin indicates that *T. reesei* can use this complex polyaromatic as a source of nutrients to complete its lifecycle. Extensive studies on lignin modification were carried out using basidiomycetes while ascomycetes have been described for their specificity towards plant polysaccharides. *T. reesei* for example, is widely used for the commercial production of hemicellulases and cellulases, however, very little is known about its potential role in lignin modification. A number of studies have reported the growth and activity of ascomycetes on lignin-rich wood substrates [27,63,64,65,66]. The fungus *Phoma herbarum* utilized natural LignoBoost kraft lignin, lignosulphonate, lignin extracted from spruce wood and synthetic lignin as a sole carbon source [67,68,69]. On the other hand, only two studies involved *Trichoderma* strains including *T. viride*, and eighteen *Trichoderma* strains isolated from lignocellulose composts. The growth of *T. viride* on basal mineral salt agar plates supplemented with 0.5 % lignin have been qualitatively followed and the fungi showed mycelial growth as bluish-green patches under this condition; however, the source of the lignin substrate used was not specified [70]. The ligninolytic activity of *Trichoderma* strains isolated from lignocellulose composts have been evaluated by measuring the decolorization effect, the enzymatic activity and the concentration of released phenolic compounds during growth on dark post-industrial lignin (wastewater originating from the pulp and paper industry) [71]. The release of greater amounts of phenolics during growth on this substrate was associated with high superoxide dismutase-like and horseradish-like activities. To our knowledge, this study represents the first report confirming that *T. reesei* can grow and utilize a technical soda lignin.

### 3.2. Characterization of Residual Lignin

The analysis of released soluble lignin monomers and oligomers after 14 days growth of *T. reesei* revealed changes in the composition of this substrate, characterized by a significant decrease in soluble high molar mass compounds (elution time < 18 min) (Figure 1A). LC–MS analysis confirmed that the small phenolic molecules, vanillin, *p*-coumaric acid and syringaldehyde almost completely disappeared in the fungal culture (Appendix A). This was paralleled by an increase in the lignin-derived monomers *p*-OH-benzaldehyde, syringic acid, and acetosyringone. The formation of acid compounds was probably due to the enzymatic oxidative conversion of phenolic lignin side chains. The presence of benzaldehyde, on the other hand, indicates that ligninolysis is happening through Cα-Cβ cleavage reactions [72]. 

HPSEC analysis of the insoluble lignin fractions indicated the consumption of almost all the phenolic monomers (Figure 1B). Structural analysis of this fraction revealed a decrease of total free phenol content and a 67% decrease in the thioacidolysis yield, suggesting that *T. reesei* modified the structure of lignin by cleaving the β-O-4 linkages (Appendix A). These structural features have also been observed during growth of *Podospora anserina* on lignin-rich wood substrates [26,27]. A shift in the first peak representing oligomers to a shorter retention was observed after fungal treatment and the polymers peak accumulated into the higher area, indicating that polymerization reactions are also taking place.

### 3.3. Proteomic Analysis

To link the observed structural changes to potential enzymatic activities, proteomic analysis of the secreted proteins during growth on lignin was performed. Previously, the production of ligninolytic enzymes by ascomycetes was mainly demonstrated by the discoloration of structurally similar dyes and measuring enzymatic activities (lignin peroxidase, manganese peroxidase and laccase) in the culture supernatants [27,73,74]. In this study, we report for the first time the enzymes produced by *T. reesei* during growth on technical lignin. In total, 245 proteins were identified in the secretome during growth on lignin alone (Appendix A) and the highest number of proteins was detected on day 7 of growth (Figure 2). The induced CAZymes (95 proteins) belonged predominantly to the glycoside hydrolases (GH) families, with 14 GHs being detected exclusively in the absence of maltose. *T. reesei* also secreted two polysaccharide lyases and five carbohydrate esterases (CE), including CE1, CE5, CE9 and CE16.

Out of the 31 predicted auxiliary activity CAZymes in the genome of *T. reesei* [38,75,76], only 5 were detected in the secretome. Interestingly, all induced AA enzymes are hydrogen peroxide-producers. The role of these enzymes in lignin degradation was previously limited to the generation of hydrogen peroxide for the activity of peroxidases. However, the lack of lignin peroxidase- and manganese peroxidase-encoding genes in *T. reesei* and the detection of structural changes in lignin during growth suggest a direct role of enzymes of this type in lignin modification by this fungus. The accumulation of hydrogen peroxide leads to the generation of highly reactive hydroxyl free radicals which are able to depolymerize lignin and expose functional groups that can be in turn recognized and oxidized by these enzymes. However, H_2_O_2_ accumulation can also have a detrimental effect on fungal cells and enzymatic activity [77]. Interestingly, one peroxidase/catalase (JGI ID 70803) and one haem-peroxidase enzymes (JGI ID 73523) were detected in the secretome. The proteins have a signal peptide and were more abundant at day 7 of growth on lignin. The peroxidase/catalase protein identified in *T. reesei* secretome shared 58 % sequence identity (Clustal Omega; Ref. [78]) with a previously described peroxidase/catalase from *Podospora anserina* (B2ASU5) which have been found to be required during growth and utilization of complex biomass like wood shavings and lignin [25]. This enzyme has also been found to play a major role in detoxifying H_2_O_2_ during vegetative growth. The secretion of these enzymes by *T. reesei* counterbalance the abundant presence of H_2_O_2_-producing AA enzymes and confers resistance to the oxidative damage during growth on lignin.

Secreted AAs included one putative aryl-alcohol oxidase (AAO)/glucose-1-oxidase (GO) (AA3_2) and one putative alcohol oxidase (AA3_3). These enzymes belong to the glucose-methanol-choline oxidase/dehydrogenase protein superfamily and are characterized by the presence of flavin adenine dinucleotide as a cofactor [79]. It has been shown that the enzymatic demethylation of lignin by fungi results in the accumulation of methanol which then induces the production of alcohol oxidases [80]. In addition, AAO is potentially implicated in lignin modification by acting on lignin-derived phenolic aromatic aldehydes and acids [81,82,83]. Enzyme belonging to the AA3 family were previously identified in the secretomes of several basidiomycetes [84,85] and ascomycetes [26,86] during growth on lignin-rich substrates. 

Another detected AA enzyme during growth on lignin was 1,4-benzoquinone reductase (AA6). This enzyme catalyzes the reduction reactions of extracellular quinones producing hydroquinones that can in turn oxidize oxygen producing reactive oxygen species [87]. AA6 enzymes have been found to be induced by vanillin, vanillate, and quinones in the basidiomycete *Phanerochaete chrysosporium* [88] and are involved in the metabolism of low-molecular weight lignin fragments [89]. 

A lytic polysaccharide monooxygenase (LPMO, AA9) was also detected in the secretome at day 7 of growth. LPMO AA9s are known for their activity on cellulose microfibrils, and they have been identified in the secretomes of both ascomycetes and basidiomycetes during growth on lignocellulosic biomass [90]. Their role in lignin modification was very recently brought into light when the H_2_O_2_ generated by LPMO was favorably used for lignin oxidation by lignin-degrading peroxidases [91]. Interestingly, LPMO was exclusively detected in the condition containing technical lignin alone. The same finding was previously reported with *Pycnoporus sanguineus* and *Leiotrametes menziesii* [42], suggesting a role of these enzymes not only in cellulose deconstruction, but also directly in lignin modification. Alternatively, it could be that lignin-derived molecules are involved in the regulation of LPMO secretion on lignocellulosic biomass. It has also been shown that lignin polymer can act as the electron donor of LPMOs during cellulose oxidation significantly improving the enzymatic hydrolysis of cellulose [92].

During growth on lignin, *T. reesei* also secreted CRO (AA5). The fungus possesses only one gene encoding for an AA5 protein, and the enzyme was detected as early as day 3 of growth and remained detectable until day 14. The detection of this enzyme in the secretomes of *T. reesei* QM6a and three of its mutants have been previously reported during growth on cellulose fibrous as a major carbon source [93]. AA5 enzymes share similar tertiary structure but have very different catalytic properties and low overall sequence similarity [36,39,58]. They include AA5_1 (GLOX and other CRO), and AA5_2 enzymes (GalOx, raffinose oxidase and alcohol oxidases) and act on a broad range of substrates including alcohol- and aldehyde-containing molecules. Interestingly a large group of CRO fall within the group of “other copper radical oxidases” annotated as CRO1 to CRO6 and which functions are still unknown [35,94]. The secretion of these enzymes was previously reported in *P. chrysosporium* during growth on lignin and thin wood sections [94,95,96] and in *Pycnoporus sanguineus* and *L. menziesii* during growth on the same technical lignin used in the current study [42]. Previously reported CRO enzymes induced on lignin belonged to CRO1 (*P. chrysosporium*, *Pycnoporus sanguineus*, *L. meziesii* and *Polyporus brumalis*) CRO2 (*Pycnoporus sanguineus*, *L. meziesii* and *Polyporus brumalis*), CRO4 (*P. chrysosporium*) and CRO5 (*Pycnoporus sanguineus*, *L. meziesii* and *Polyporus brumalis)* groups [42,96]. However, the catalytic properties and the role of these enzymes in lignin modification have never been investigated.

### 3.4. Phylogenetic Analysis

As the detected AA5 was the first reported from ascomycete to be induced on technical lignin, the enzyme was targeted for heterologous production and in vitro characterization in the aim of elucidating its role in lignin modification. The enzyme is referred to as lignin oxidase (*Tr*LOx). The phylogenetic tree grouping characterized and uncharacterized AA5_1 and AA5_2 enzymes from Basidiomycetes and Ascomycetes showed a clear distinction between the two subfamilies (Figure 3). The AA5_1 subfamily further separates into two subgroups with the first one containing mainly AA5_1 from Ascomycetes including *Tr*LOx and the second AA5_1 from Basidiomycetes with few exceptions. *Tr*LOx clusters with its homolog from *T. virens* which has been recently described for its role in normal hyphal growth and morphology [97]. However, the substrate profile of GLX1 from *T. virens* has not been investigated. In addition, none of the previously produced and characterized AA5 enzymes fell in the same subgroup containing *Tr*LOx, suggesting possible divergent properties of this enzyme.

To identify variations in the amino acids previously described to be involved in catalysis such as copper coordination, and substrate recognition, *Tr*LOx protein sequence was aligned with functionally characterized AA5_1 and AA5_2 enzymes (Appendix A). The alignments showed conserved copper coordinating residues (Cys252, Tyr311, Tyr533, His534 and His618). *Tr*LOx shared higher similarity with characterized GLOX for residues known to be involved in catalysis and substrate recognition of D-galactose in AA5_2 enzymes [40]. 

*Tr*LOx encoding sequence contained five WSC domains at the N-terminal end (Appendix A). The function of this domain remains poorly studied. A recent study has shown no contribution of WSC domain to the catalytic activity of an alcohol oxidase from *Pyricularia oryzae* [98]. Instead, the authors have demonstrated a role of this domain in binding the enzyme to plant and/or fungal cell wall via xylans and fungal chitin/β-1,3-glucan. Furthermore, Crutcher et al. (2019) have shown that GLX1 from *T. virens* binds to chitin but not cellulose, lignin or peptidoglycan. Previously, a role in fungal responses to cell wall disruption, oxidation, high osmolarity, and metals stress have also been presented [99]. GLX1-silenced *T. virens* mutants have shown reduced sporulation, hydrophobicity and a loss of growth directionality in the hyphal tips due to the reduced production of H_2_O_2_ [97]. Taking these results into consideration, it is possible that the role of the WSC domain in *Tr*LOx is to resist to the stress induced by the presence of technical lignin in the culture. 

The predicted molecular structure of *Tr*LOx generated with copper oxidase from *Colletotrichum graminicola* (PDB 6RYX) as template shows the conserved residues coordinating the catalytic copper ion H960, H1044, Y959, and Y737 that forms the characteristic thioether linkage to C678 (Appendix A) [39,100,101].

### 3.5. Production of Active TrLOx

The best-producing *P. pastoris* transformants were selected and used for the large-scale production of *Tr*LOx. IMAC purification yielded 32.5 mg/L of the recombinant protein. On SDS-PAGE, *Tr*LOx showed a homogeneous single band at ≈150 kDa, 34.2 kDa larger than the theoretical molecular weight of the protein, suggesting the presence of glycosylation (Appendix A). This was further supported by the prediction of 8 N-glycosylation and 34 O-glycosylation sites using NetNGlyc 1.0 and NetOGlyc 4.0, respectively [102,103]. 

Enzymatic activity was assayed by following H_2_O_2_ production in a coupled reaction with HRP and ABTS. This method was previously used to measure the activity of glyoxal oxidases and galactose oxidases [36,58,59]. The specificity of *Tr*LOx towards seventy different substrates was tested and the enzyme showed a broad range of activity (Table 1). 

The highest activity was detected on dihydroxyacetone and the screening revealed a weak activity towards glyoxylic acid, the best previously reported substrate for GLOX from *Pycnoporus cinnabarinus* [36]. On the other hand, the activity on methylglyoxal, the best substrate for GLOX from *Phanerochaete chrysosporium*, was higher [58]. High activity on benzyl alcohols which are common structures in lignin was also observed. Furfuryl alcohol and veratryl alcohol were for example very competent substrates. Unlike other characterized GLOX enzymes, *Tr*LOx showed weak activity on carbohydrates. These substrates are readily oxidized by the related enzyme, GalOx [104]. However, *Tr*LOx was different from AA5_2 enzymes by showing higher activity on monosaccharides, notably xylose and no detectable activity on polysaccharides [40,105]. In addition, AA5_2 have very weak activity towards aliphatic, aromatic, phenolic and heterocyclic compounds compared to *Tr*LOx [106]. Interestingly vanillin, which was found to decrease in the cultures of *T. reesei* on technical lignin was also oxidized by this enzyme. 

The higher activity observed with dihydroxyacetone and methylglyoxal suggests that, in contrast to GLOX, the enzyme favors hydroxyl compounds and recognizes to a lesser extent aldehydes. In addition, polyols were very poor substrates which was reflected by the drastic difference in the detected activities on glycerol compared to dihydroxyacetone. Previously, dihydroxyacetone was found to be more efficiently oxidized by AA5_2 enzymes compared to the standard substrate galactose [107,108]. The substrate has been found to replace the water molecule at the equatorial site of the copper center [109]. Dihydroxyacetone can be hydrated generating gem-diol which have been found to be favored substrates for AA5_2 enzyme [110]. 

The preferential oxidation of alcohol to their corresponding aldehydes was further confirmed by product analysis of the oxidations of 5-hydroxymethyl furfural (HMF) derivatives (Appendix A). HMF is particularly important as by-product from biorefineries that can be transformed into high-value building blocks for green chemistry. The products of the oxidation of HMF and HMFCA by *Tr*LOx were 2,5-furandicarboxaldehyde (DFF) and 5-formyl-2-furan carboxylic acid (FFCA), respectively, and no further conversion of DFF and FFCA was observed. *Tr*LOx clearly favors the oxidation of the alcohol group of these substrates, further highlighting the difference with other AA5_1 enzymes such as GLOX [34]. Our results suggest that enzymes originating from the same subfamily AA5_1, namely *Tr*LOx and GLOX, possess orthogonal activities and are potential candidates for one-pot conversion of HMF to furandicarboxylic acid (FDCA). The oxidation of the alcohol groups of HMF and HMFCA was previously reported for a newly characterized AA5_2 enzyme exhibiting aryl alcohol oxidase activity [101]. 

Kinetic analyses using preferred substrates revealed a classical Michaelian trend except for galactose and glycerol which appear as strictly linear plots (Appendix A). The highest catalytic efficiency was obtained on dihydroxyacetone (*K*_cat_/*K*_m_ = 5.89 s^−1^mM^−1^). However, *Tr*LOx showed higher affinity to 2-phenylethanol (*K*_m_ = 12.32 mM), methylglyoxal (*K*_m_ = 11.03 mM) and furfuryl alcohol (*K*_m_ = 6.72 mM) compared to dihydroxyacetone (*K*_m_ = 24.16 mM) under the used conditions (Table 2). 

H_2_O_2_ generation correlated with the measured activities on the selected substrates with the highest rate being detected in the reaction on dihydroxyacetone (Appendix A). The detected fluorescence peaked and then declined during the course of the reaction. This effect has been found to be triggered by different factors including the fluorescent product resorufin being itself a substrate for HRP, substrate inhibition and inactivation of HRP at high concentrations of hydrogen peroxide leading to a bell-shaped titration curve with hydrogen peroxide concentration range of 0.01 to 600 µM [111] as observed in this study, notably with dihydroxyacetone.

### 3.6. Biochemical Properties

The effect of different metal ions, metal chelator and sodium azide on the stability and activity of *Tr*LOx was investigated after treating the enzyme with 10 mM of each molecule for 4 and 24 h (Figure 4A). Treatment with 10 mM Cu^2+^ for 4 h resulted in 4.5 times increase in enzymatic activity. Previously, adding copper sulfate to shake flask cultivations was found to double GalOx activity in the cultivation medium [112]. However, unlike *Tr*LOx, copper treatment of purified AA5_2 enzymes did not increase their specific activity [105,112]. Recombinant AA5 enzymes have been found to contain a mixture of fully coordinated proteins and apo-enzymes lacking copper [113]. The increase in activity for *Tr*LOx suggests that the purified enzyme is not fully coordinated by Cu^2+^ ions from the culture media. No significant effect was observed with any of the other tested molecules after 4 h of incubation. The enzyme activity was reduced by 70% and 60% after 24 h of incubation with Co^2+^ and Fe^2+^, respectively. The loss in activity observed with the copper chelator EDTA (56%) was comparable to previously published data obtained on a recombinant GalOx from *Fusarium sambucinum* [59]. However, unlike this enzyme which was completely inactive after incubation with sodium azide (5 mM; 5 min), *Tr*LOx was significantly more stable retaining 60% of its activity after 24 h of incubation. Azide is thought to inhibit GalOx by replacing the water molecule in the catalytic center of the enzyme and blocking the substrate’s binding site [114]. 

The stability of *Tr*LOx over time in the presence of increasing concentrations of H_2_O_2_ was also investigated (Figure 4B). The enzyme was stable after incubation for two hours in up to 8 mM of H_2_O_2_. A concentration-dependent decrease in residual activity was observed with increased incubation time and the enzyme was completely inactive after 24 h with 10 mM H_2_O_2_. The inhibition of the AA5_1 enzyme GLOX by exogenous H_2_O_2_ has been previously reported. *Tr*LOx was significantly more stable than GLOX from *P. chrysosporium* (25% residual activity directly after adding 2.1 mM exogenous H_2_O_2_ at pH 6) and GLOX2 and GLOX3 from *P. cinnabarinus* (less than 10% residual activity after 24 h of incubation with 8 mM H_2_O_2_) [34,41].

Dihydroxyacetone was used to determine the optimum pH and temperature for the activity of *Tr*LOx. The highest activity was detected at pH 6.5 (Figure 5A) and 45 °C (Figure 5B).

*Tr*LOx was very sensitive to acidic pH, the enzyme showed high stability in a pH range of 4–8 (Figure 6A). *Tr*LOx retained 34% of its activity after incubation at 40 °C for 8 h and was stable at 30 °C (Figure 6B). No activity was detected when the enzyme was exposed to temperatures above 60 °C. The conditions under which *Tr*LOx showed highest activity were comparable to previously characterized *P. cinnabarinus* GLOX enzymes (pH 6 and 50 °C) [36]. In addition, *Tr*LOx was more stable at 30 °C and less stable at temperatures exceeding 50 °C compared to *P. cinnabarinus* GLOX. Similarly, *P. chrysosporium* GLOX has shown highest activity at pH 6 and a sharp decrease in activity at pH below 5.5 [115]. However, *P. chrysosporium* GLOX has completely lost activity at pH 7.5 and was not active in phosphate buffer. In comparison, AA5_2 enzymes have shown higher activity at pH 7–8 and the activity was halved at pH 6 [40,101,105]. Similarly to *Tr*LOx, characterized AA5_2 enzymes have also shown reduced thermal stability. 

### 3.7. Does TrLOx Modify Lignin?

Previous studies have presumed a role of CROs in lignin modification either directly or through the generation of hydrogen peroxide. Crutcher et al. (2019) have recently ruled out this hypothesis by showing that GLX1 from *T. virens* was not implicated in the breakdown of lignin as the enzyme was unable to bind lignin. However, the direct activity of CROs on this substrate and its derivatives and the resulting structural modifications have never been explored. 

In this paper, the activity of *Tr*LOx was investigated for the first time on syringol, syringaldehyde, and three lignin dimers: BMA32 (SS βO4), AMA170 (SS syringaresinol) and AMA181 (GG β5). The tested dimers represent the major structural units in the polymer. *Tr*LOx activity was followed by LC-MS and HPSEC. The HPSEC analysis showed that *Tr*LOx had no activity on AMA170 (SS ββ), whereas it had an activity on AMA181 (GG β5) and BMA32 (SS βO4) (Figure 7), showing oligomers with polymerization degree (PD) up to 10. The LC-MS analysis confirmed that the observed reactions are oligomerization and oxidation of functional groups (alcohols into aldehydes). Concerning BMA32, though the main products formed are dimers and trimers of the initial substrate (PD 4 and 6), the detection of some oligomers by LC-MS with higher degree of polymerization (observed with the SS βO4 substrate) suggest that some bonds’ cleavage might occur during the treatment or that the monomers present as traces in the initial material are involved in the oligomerization. It seems that the reactivity of the systems depends on the dimer type, which indicates enzymes specificities towards the functional groups present in these molecules. The activity of *Tr*LOx was also compared to GLOX2 from *Pycnoporus cinnabarinus* for which no activity was noticed towards the three dimers compared to *Tr*LOx. 

In nature, plants incorporate numerous lignin monomers to form lignin by oxidation and coupling of monolignol radicals with the radicals on the phenolic ends of the lignin polymer. This reaction is initiated by the action of plant oxidative enzymes such as O_2_-dependent laccases and H_2_O_2_-dependent peroxidases and it serves not only for the production of lignin polymers but also to link this structure to the polysaccharides in the cell wall. Lignin polymerization was also previously reported in fungal cultures on lignin [73] and it was associated with the oxidation of lignin phenoxy radicals by secreted fungal laccases in the cultures [74]. The observed oxidation of the phenolic moieties of syringol to form syringol dimers shows that *Tr*LOx polymerizes lignin following the same mechanism (Appendix A). The enzyme seems to favor the oxidation of the hydroxy moiety of the hydroxymethyl group in syringyl alcohol (Appendix A). In the case of lignin dimers, it is difficult to determine the exact coupling sites. However, the absence of activity on AMA170 suggest that the coupling most likely occurs between the oxidized alcohol groups at the hydroxypropenyl side chain and at the C4 position of the syringyl and guaiacyl units of BMA32 and AMA181, respectively (Figure 7). The oxidized new radical will then react either with new dimers or oligomers formed in the reaction resulting in oligomers with PD up to 10. This is further supported by the measured activity on cinnamyl alcohol and vanillin (Table 1). Regarding in-vivo conditions, polymerization is thought to be prevented by the presence of cooperative enzymes acting as quenchers to radical species. Several flavin adenine dinucleotide (FAD)-containing enzymes have been described to reduce phenoxy radicals and prevent lignin re-polymerization, including glucose oxidase [116], veratryl alcohol oxidase [117], cellobiose dehydrogenase [118], and pyranose-2-oxidase [119]. The latter not only prevented repolymerization, but also showed depolymerizing activity on lignosulfonate. This suggests that polymerizing enzymes such as laccase and *Tr*LOx might act by activating lignin hydroxyl radicals for degradation by other secreted enzymes. Interestingly, one aryl-alcohol oxidase (AAO)/glucose-1-oxidase (GO) and one alcohol oxidase (AA3_3) were co-secreted with *Tr*LOx in the cultures of *T. reesei* on technical soda lignin, further supporting this hypothesis. In addition, *T. reesei* secreted a 1,4-benzoquinone reductase which might be involved in the reduction of reactive phenoxy radicals and/or quinones formed during lignin breakdown converting them into more stable molecules. The effect of combined AA6 and lignin peroxidase on the depolymerization of technical lignins was very recently investigated and the study revealed that AA6 can to some extent limit the lignin re-polymerization by lignin peroxidase [120]. Further studies investigating the synergistic activity of *Tr*LOx and the above-mentioned enzymes on lignin molecules are required to elucidate the mechanism behind lignin modification. 

The results in this study also introduced for the first time the potential of *Tr*LOx for the synthesis of new homomolecular or heteromolecular molecules. Current applications in the biopolymer field include dye synthesis [121], cotton fibers functionalization [122], and hydrogels production for drug delivery and water treatment [123]. Such biopolymers are perfect examples of sustainable biotechnological application. 

## Figures and Tables

**Figure 1 jof-07-00643-f001:**
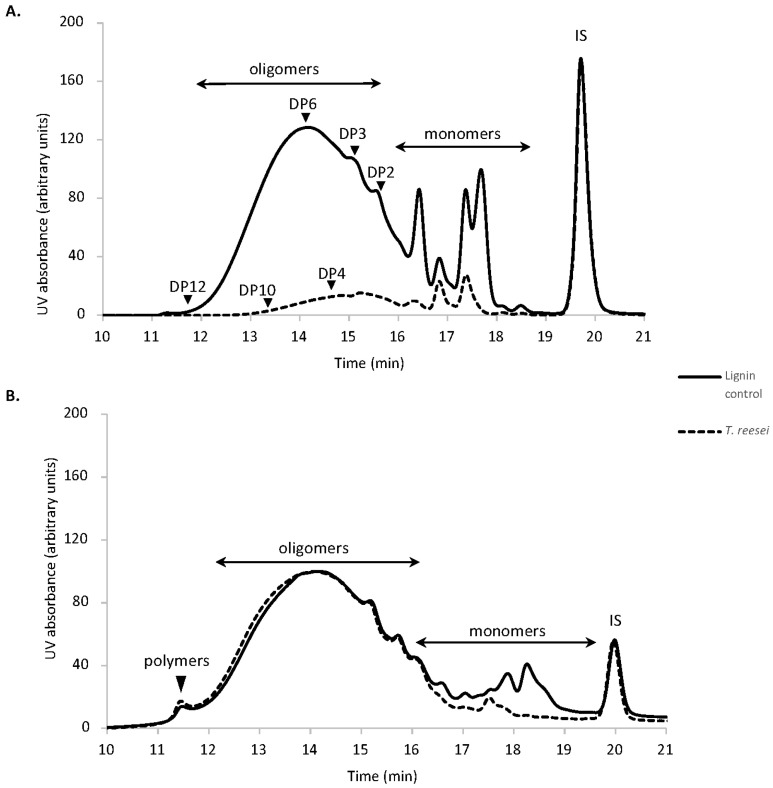
HPSEC analysis of residual lignin substrate. (**A**) water-soluble residual lignin fraction and (**B**) water insoluble residual lignin fraction. HPLC was performed on the ethyl acetate extracts of the culture supernatants and pellets recovered from soda lignin after 14-day incubation with *T. reesei*; chromatograms normalized on IS. Eluent tetrahydrofuran (THF), 1 mL/min; detection at 280 nm; 100 Å PL-gel column (Polymer Laboratories, 5 μm, 600 mm × 7.5 mm). The lignin control was dissolved in the culture media in the absence of fungi and incubated under the same conditions. DP: degree of polymerization, IS: internal standard is toluene.

**Figure 2 jof-07-00643-f002:**
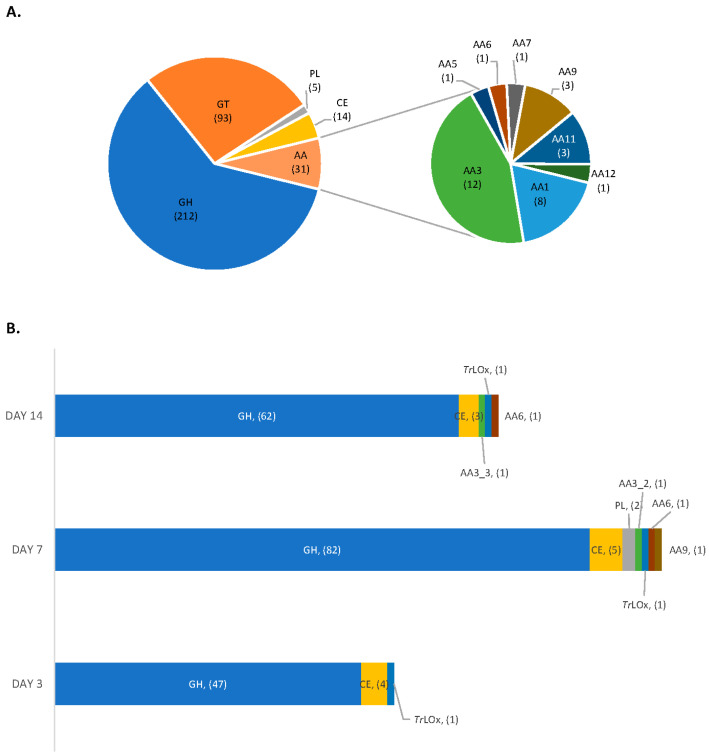
Distribution of CAZymes in genome and secretome of *T. reesei*. (**A**) genome and (**B**) secretome of *T. reesei* on lignin at days 3, 7 and 14 of growth. AA, auxiliary activity enzymes; GH, glycoside hydrolases; GT, glycosyl transferases; PL, polysaccharide lyases; CE, carbohydrate esterases; AA1, multicopper oxidases; AA2, class II lignin-modifying peroxidases; AA3, glucose-methanol-choline (GMC) oxidoreductases; AA3_2, aryl alcohol oxidase and glucose 1-oxidase; AA3_3, alcohol oxidase; AA5, copper radical oxidases; AA6, 1,4-benzoquinone reductase; AA7, glucooligosaccharide oxidases; AA9, AA11 and AA14 lytic polysaccharide monooxygenases; AA12, pyrroloquinoline quinone-dependent oxidoreductase; *Tr*LOx, lignin copper oxidase.

**Figure 3 jof-07-00643-f003:**
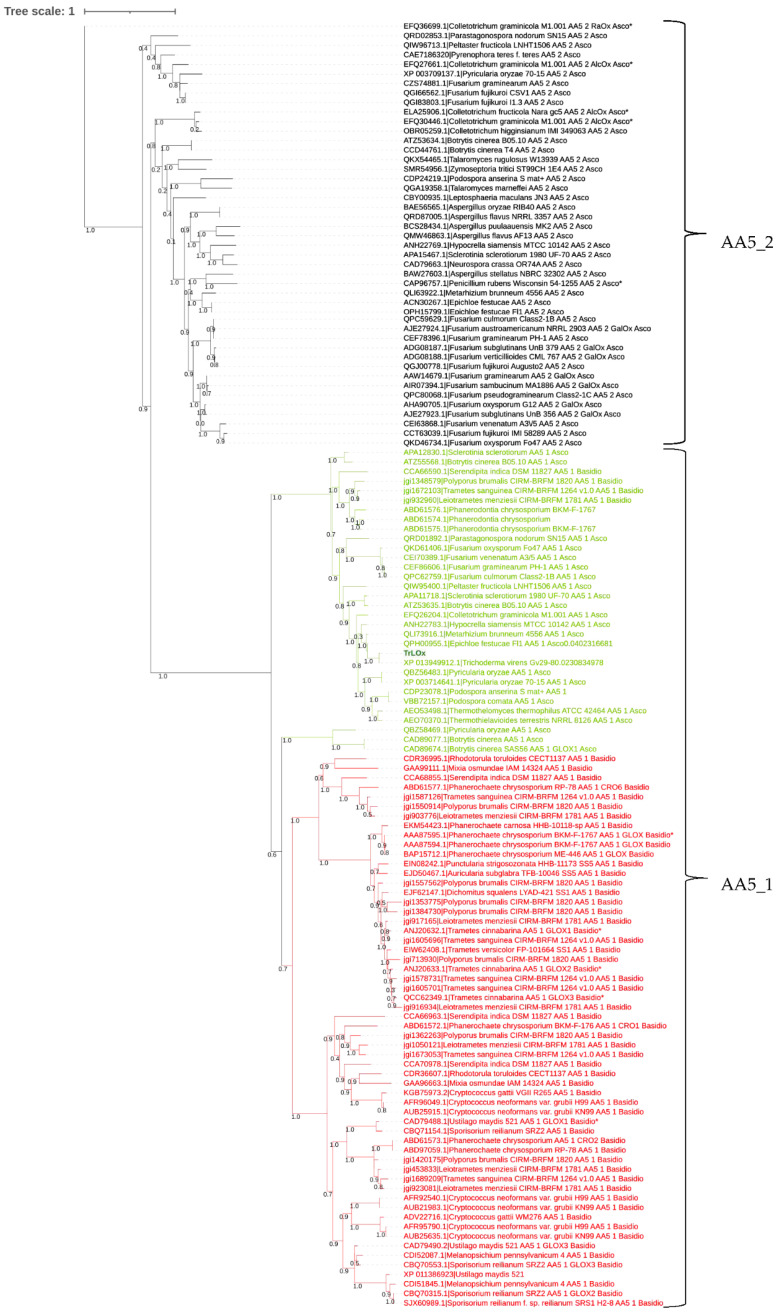
Phylogenetic tree of AA5 proteins. GenBank and JGI identifiers are given for all sequences. Subfamilies AA5_1 (green: from ascomycetes; red: from basidiomycetes) and AA5_2 (black) are indicated. Functionally characterized enzymes are indicated with a star (*). Multiple sequence alignment was performed using MAFFT tool and the tree was constructed using NGphylogeny and PhyML. The branches represent evolutionary changes measured in unit of genetic divergence. The bar at the top provides the scale of the branch length.

**Figure 4 jof-07-00643-f004:**
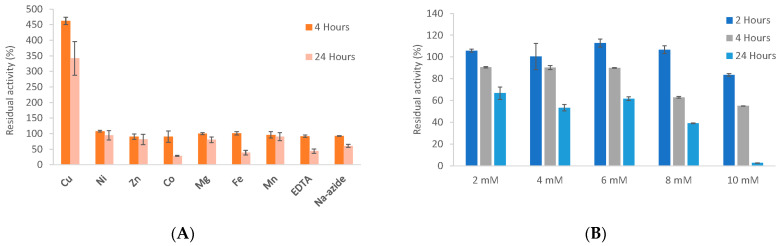
Enzyme stability in the presence of metals, inhibitors and H_2_O_2_. (**A**) Effect of 10 mM of Cu, Ni, Zn, Co, Mg, Fe, Mn, EDTA and sodium azide on the enzymatic activity of *Tr*LOx after incubation for 4 (dark orange) and 24 h (light orange); (**B**) Enzyme stability in the presence of increasing concentrations of hydrogen peroxide over an incubation period of 2, 4 and 24 h. Activities were calculated as a relative percentage of the activity in the absence of any molecule and reported as mean. Standard deviations are presented by error bars.

**Figure 5 jof-07-00643-f005:**
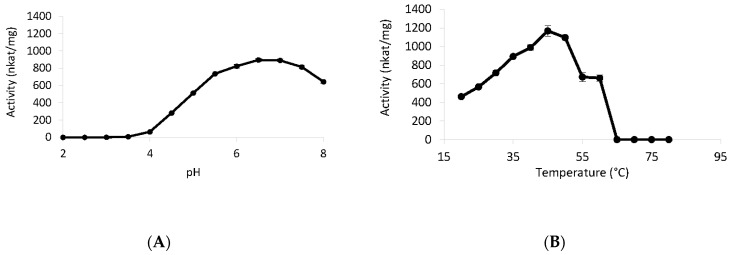
Optimum pH and temperature for the activity of *Tr*LOx. (**A**) Activity in Sodium tartrate (pH 2–6) and sodium phosphate (pH 6–8) buffers; (**B**) Measured enzymatic activity at different temperatures. Standard deviations are presented by error bars.

**Figure 6 jof-07-00643-f006:**
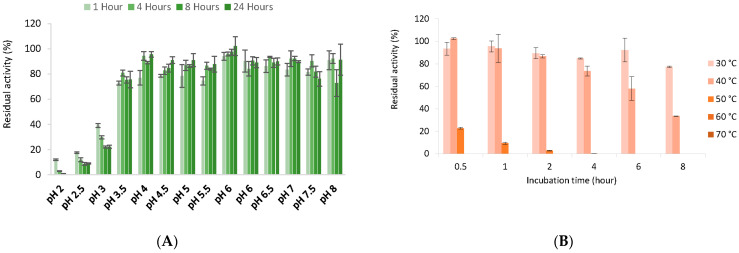
Enzyme stability under different pH and temperatures. (**A**) Stability after incubation for different time periods at pH 2–8; (**B**) Stability after incubation for different time periods at 30–70 °C. Measured enzymatic activity at different temperatures. Activities were calculated as a relative percentage of the activity before incubation. Mean values are reported, and standard deviations are presented by error bars.

**Figure 7 jof-07-00643-f007:**
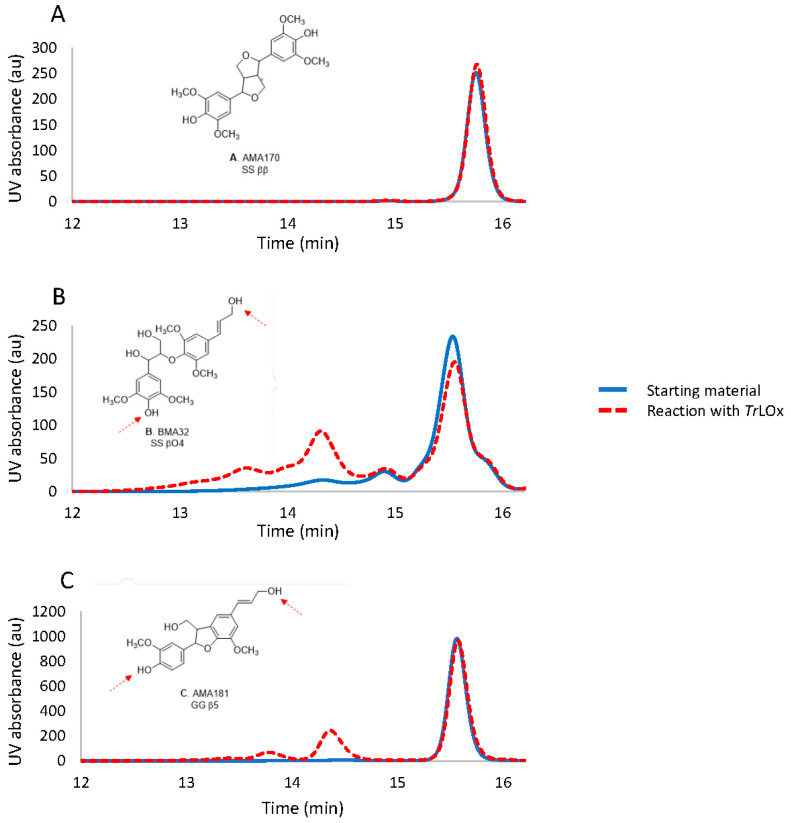
HPSEC-PDA (size-exclusion chromatography coupled to a photodiode array detector, absorbance 280 nm) profiles of the starting dimer and their reactions with *Tr*LOx, reaction with (**A**) AMA170, (**B**) BMA32 (**C**) AMA181; starting material (blue); reaction with *Tr*LOx (red). The red arrows indicate the potential coupling site by the action of *Tr*LOx.

**Table 1 jof-07-00643-t001:** Substrate specificity of *Tr*LOx.

	Substrate	Activity (nkat/mg)
Phenolic	Syringyl alcohol	0.44 ± 0.01
	Caffeic acid methyl ester	0.24 ± 0.01
	p-coumaric acid methyl ester	0.18 ± 0.01
	Vanillin	0.12 ± 0.01
	Acetovanillone	0.10 ± 0.02
	Homovanillic acid	0.08 ± 0.01
	Sinapic acid	0.05 ± 0.001
	Isovanillin	0.05 ± 0.01
	4-Ethylphenol	0.04 ± 0.01
	Vanillyl alcohol	0.03 ± 0.001
	p-coumaryl alcohol	0.02 ± 0.01
	4-vinylguaiacol	nd
	p-hydroxybenzaldehyde	nd
	Syringol	nd
	Syringaldehyde	nd
	p-coumaric acid	nd
	Syringic acid	nd
	Sinapic acid methyl ester	nd
	Homovanillyl alcohol	nd
	Eugenol	nd
	Vanillyl acetone	nd
	Ferulic acid	nd
	Ferulic acid methyl ester	nd
	Coniferyl alcohol	nd
	Guaiacol	nd
	4-ethylguaiacol	nd
	Vanillic acid	nd
	Caffeic acid	nd
	Chlorogenic acid	nd
	Phenol	nd
Carbohydrate	Xylose	1.73 ± 1.31
	Galactose	1.27 ± 0.37
	Glucose	0.64 ± 0.01
	Raffinose	0.44 ± 0.01
	Cellobiose	nd
	Fructose	nd
	Maltose	nd
	Maltotriose	nd
Furan	Furfuryl alcohol	12.13 ± 0.29
	5-Hydroxymethylfurfural	1.30 ± 0.36
	5-Hydroxymethyl-2-furancarboxylic acid	0.43 ± 0.02
	5-formylfuran-2-carboxylic Acid	0.30 ± 0.08
	Furfural	nd
	Furan-2,5-dicarbaldehyde	nd
Alcohol	Veratryl alcohol	5.49 ± 1.36
	2-phenylethanol	4.43 ± 0.22
	1-phenyl-3-propanol	1.92 ± 0.04
	Glycerol	1.91 ± 0.09
	Cinnamyl alcohol	0.99 ± 0.01
	1-phenylethanol	0.81 ± 0.06
	Sorbitol	nd
	(±)-2-octanol	nd
	Methanol	nd
	Diethanolamine	nd
	Triethanolamine	nd
Aldehyde, Ketone, Carboxylic acid	Dihydroxyacetone	268.86 ± 5.71
	Methylglyoxal	26.30 ± 1.15
	Glyoxylic acid	1.73 ± 0.16
	Veratric acid	0.93 ± 0.02
	Formaldehyde	0.76 ± 0.11
	Glyceraldehyde	0.64 ± 0.02
	Glyoxal	0.51 ± 0.03
	Acetaldehyde	nd
	Veratraldehyde	nd
	Phenyl glyoxylic acid	nd
	Acetone	nd
	Acetophenone	nd
Quinone	Benzoquinone (quinone)	nd
Aniline	Ortho-anisidine (aniline)	nd
Lactone	D-Xylono-1,4-lactone	nd

nd: not detected.

**Table 2 jof-07-00643-t002:** Kinetic parameters of *Tr*LOx with different substrates. Standard deviations are presented as plus–minus values.

Substrate	*V*_max_ (nkat/mg)	*K*_m_ (mM)	*K*_cat_ (s^−1^)	*K*_cat_/*K*_m_ (s^−1^ mM^−1^)
Dihydroxyacetone	2455.81 ± 56.25	24.16 ± 1.44	142.19	5.89
Methylglyoxal	46.84 ± 6.39	11.03 ± 2.84	1.08	0.10
Furfuryl alcohol	13.28 ± 1.56	6.72 ± 1.91	0.31	0.05
2-Phenylethanol	11.23 ± 1.53	12.32 ± 5.18	0.26	0.02

## Data Availability

The data presented in this study are available on request from the corresponding author. The data are not publicly available due to confidentiality.

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
