# Peer review of "A Putative Lignin Copper Oxidase from Trichoderma reesei"

_jof, 2021, doi:10.3390/jof7080643_

Round 1

Reviewer 1 Report

In this publication, detailed biochemical characteristics of lignin copper oxidase from Trichoderma reesei is described. This publication seems to be within the scope of journal. However it needs several corrections to be more acceptable for publication.

  1. Has the authors tried to isolate the products of the enzymatic reaction? Detailed NMR analysis would allow to clearly determine which dimer is the product of the TrLOx reaction with syringol.
  2. In section 2.9 please add information about method of converting absorbance to mkat/mg (Table 1).
  3. In section 2.3 please add detailed information about GC-MS analysis: type of detector, temperature of detector, temperature of injector, used temperature programme, carrier gas with flow rate, split ratio, ionization energy, mass scan range.
  4. In line 149, the authors should explain what is control.
  5. Was the analysis performed isocratic with 2.5 mM sulfuric acid as eluent?
  6. In line 211 please explain what YNB means. In line 216 please explain what PTM4 means.
  7. In line 90, it should be “69-94%” instead of “69-4%”
  8. In line 92, name of fungus genus should be written in italic.
  9. In line 227 it should be “mL” instead of “ml”. Please correct evident mistake.
  10. It should be “g/L” instead of “L-1”. Please correct evident mistake in whole manuscript.
  11. In supplementary materials table title should be above Table S1.
  12. In supplementary materials, there is lack of explanation, what every used colour mean in Fig S5.
  13. The bibliography has been duplicated. Please remove the redundant version.

Author Response

In this publication, detailed biochemical characteristics of lignin copper oxidase from Trichoderma reesei is described. This publication seems to be within the scope of journal. However it needs several corrections to be more acceptable for publication.

  1. Has the authors tried to isolate the products of the enzymatic reaction? Detailed NMR analysis would allow to clearly determine which dimer is the product of the TrLOx reaction with syringol.

The enzymatic reactions on syringol and syringyl alcohol were analyzed by LC-MS, however, the products of the reactions were not isolated for further analysis. Detailed NMR analysis on the reaction products of TrLOx on syringol and similar compounds would indeed be very interesting to perform in the future as we go in more depth about the function of fungal CROs, and a comparison between isoenzymes.

  1. In section 2.9 please add information about method of converting absorbance to mkat/mg (Table 1).

Lines 263-264: More information about the method of converting absorbance to nkat/mg was added.

  1. In section 2.3 please add detailed information about GC-MS analysis: type of detector, temperature of detector, temperature of injector, used temperature programme, carrier gas with flow rate, split ratio, ionization energy, mass scan range.

The GC-MS analysis was performed following the same method described in Méchin et al (2014) J Agric Food Chem(https://doi.org/10.1021/jf5019998). A detailed description of the analysis and the parameters used is provided in the cited paper. The reference was added to section 2.3 of the manuscript and the reference list was updated.

  1. In line 149, the authors should explain what is control. Was the analysis performed isocratic with 2.5 mM sulfuric acid as eluent?

Lines 151-152: The control condition consisted of the culture media containing lignin in the absence of fungi and incubated under the same conditions.

The detailed description of the method and parameters used for the HPSEC and LC-MS analyses are available in our previous paper (Daou et al (2021) J Fungi: https://doi.org/10.3390/jof7010039).

For the HP-SEC analysis:  Ethyl acetate extraction residues were dissolved in tetrahydrofuran (1 mg.mL-1) and injected into a styrene–divinylbenzene PL-gel column. Tetrahydrofuran (1 mL.min-1) was used as eluent.

For the LC-MS analysis: Ethyl acetate extraction residues were dissolved in acetonitrile (1 mg.mL-1) and injected to C18 column. Elution was performed at 5–100% vol. aqueous acetonitrile (containing 1‰ HCOOH) gradient for 30 min, and 0.4 mL.min-1 as flow rate.

Isocratic elution with 2.5 mM sulfuric acid was used for the HPLC analysis of TrLOx reaction products on HMF and HMFCA (Figure S5). This information was added to the figure lengend.

  1. In line 211 please explain what YNB means. In line 216 please explain what PTM4 means.

Section 2.6: the full names of the medium and their constituents were added.

  1. In line 90, it should be “69-94%” instead of “69-4%”

This error was corrected.

  1. In line 92, name of fungus genus should be written in italic.

This error was corrected.

  1. In line 227 it should be “mL” instead of “ml”. Please correct evident mistake.

This error was corrected throughout the manuscript.

  1. It should be “g/L” instead of “L-1”. Please correct evident mistake in whole manuscript.

This error was corrected throughout the manuscript.

  1. In supplementary materials table title should be above Table S1.

The table titles were moved to be above Table S1 and Table S2.

  1. In supplementary materials, there is lack of explanation, what every used colour mean in Fig S5.

Further expalantion was added to the legend of Figure S5.

  1. The bibliography has been duplicated. Please remove the redundant version.

The redundant version of the reference list was removed.

Reviewer 2 Report

I have enjoyed reading manuscript "A Putative Lignin Copper Oxidase from Trichoderma reesei". It is well written, factual and scientifically sound. Author put lot of work and effort into this paper and I have no criticisms whatsoever. 

Author Response

We would like to thank the Reviewer for the very kind and supportive comments made on our manuscript. We are very happy that you enjoyed reading our work.